# Empagliflozin Attenuates Myocardial Sodium and Calcium Dysregulation and Reverses Cardiac Remodeling in Streptozotocin-Induced Diabetic Rats

**DOI:** 10.3390/ijms20071680

**Published:** 2019-04-04

**Authors:** Ting-I Lee, Yao-Chang Chen, Yung-Kuo Lin, Cheng-Chih Chung, Yen-Yu Lu, Yu-Hsun Kao, Yi-Jen Chen

**Affiliations:** 1Department of General Medicine, School of Medicine, College of Medicine, Taipei Medical University, Taipei 11031, Taiwan; agleems29@gmail.com; 2Department of Internal Medicine, School of Medicine, College of Medicine, Taipei Medical University, Taipei 11031, Taiwan; yklin213@yahoo.com.tw (Y.-K.L.); michaelchung110@gmail.com (C.-C.C.); 3Division of Endocrinology and Metabolism, Department of Internal Medicine, Wan Fang Hospital, Taipei Medical University, Taipei 11696, Taiwan; 4Department of Biomedical Engineering, National Defense Medical Center, Taipei 11490, Taiwan; bme02@ndmctsgh.edu.tw; 5Cardiovascular Research Center, Wan Fang Hospital, Taipei Medical University, Taipei 11696, Taiwan; 6Division of Cardiology, Department of Internal Medicine, Sijhih Cathay General Hospital, New Taipei City 22174, Taiwan; yolu59@yahoo.com.tw; 7School of Medicine, Fu-Jen Catholic University, New Taipei City 22174, Taiwan; 8Graduate Institute of Clinical Medicine, College of Medicine, Taipei Medical University, Taipei 11031, Taiwan; yuhsunkao@gmail.com; 9Department of Medical Education and Research, Wan Fang Hospital, Taipei Medical University, Taipei 11696, Taiwan

**Keywords:** Sodium glucose co-transporter 2 inhibitor, diabetes mellitus, calcium handling, sodium regulation, cardiomyocytes

## Abstract

Diabetes mellitus (DM) has significant effects on cardiac calcium (Ca^2+^) and sodium (Na^+^) regulation. Clinical studies have shown that empagliflozin (Jardiance™) has cardiovascular benefits, however the mechanisms have not been fully elucidated. This study aimed to investigate whether empagliflozin modulates cardiac electrical activity as well as Ca^2+^/Na^+^ homeostasis in DM cardiomyopathy. Electrocardiography, echocardiography, whole-cell patch-clamp, confocal microscopic examinations, and Western blot, were performed in the ventricular myocytes of control and streptozotocin-induced DM rats, with or without empagliflozin (10 mg/kg for 4 weeks). The results showed that the control and empagliflozin-treated DM rats had smaller left ventricular end-diastolic diameters and shorter QT intervals than the DM rats. In addition, the prolonged action potential duration in the DM rats was attenuated in the empagliflozin-treated DM rats. Moreover, the DM rats had smaller sarcoplasmic reticular Ca^2+^ contents, intracellular Ca^2+^ transients, L-type Ca^2+^, reverse mode Na^+^-Ca^2+^exchanger currents, lower protein expressions of sarcoplasmic reticulum ATPase, ryanodine receptor 2 (RyR2), but higher protein expressions of phosphorylated RyR2 at serine 2808 than the control and empagliflozin-treated DM rats. The incidence and frequency of Ca^2+^ sparks, cytosolic and mitochondrial reactive oxygen species, and late Na^+^ current and Na^+^/hydrogen-exchanger currents were greater in the DM rats than in the control and empagliflozin-treated DM rats. Empagliflozin significantly changed Ca^2+^ regulation, late Na^+^ and Na^+^/hydrogen-exchanger currents and electrophysiological characteristics in DM cardiomyopathy, which may contribute to its cardioprotective benefits in DM patients.

## 1. Introduction

Cardiovascular (CV) complications are major causes of mortality in patients with both type 1 and type 2 diabetes mellitus (DM) [1,2]. DM cardiomyopathy is common in DM patients, and involves complex interactions of impaired mechanical functions and electrical abnormalities, leading to arrhythmogenesis [3,4]. Sodium glucose co-transporter (SGLT) 2 inhibitors are novel anti-DM agents that lower blood glucose levels by promoting urinary glucose excretion in patients with type 2 DM [5]. In addition, studies have also shown that SGLT2 inhibitors are effective adjunct therapy for glycemic control in patients with type 1 DM [6,7]. Empagliflozin, a highly selective SGLT2 inhibitor [8], has demonstrated CV benefits with a 38% reduction in the relative risk of CV death, and a 35% risk reduction in hospitalization for heart failure in DM patients [9]. However, the mechanisms behind the improved CV outcomes in patients receiving empagliflozin are unclear.

Homeostasis of calcium (Ca^2+^) and sodium (Na^+^) in the heart is important to appropriately regulate heart rhythm, myocardial signal transduction, energy production and respiration of the myocardium [10]. DM is associated with cardiac electrical disturbances and Ca^2+^ dysregulation [11]. In addition, both Ca^2+^ and Na^+^ transport have been shown to be altered in the hearts of patients with DM [12,13]. Previous studies have shown that different anti-DM agents may have various effects on cardiac electrical properties in animal models and in vitro [14,15,16]. The anti-DM agent, sitagliptin, a dipeptidyl peptidase-4 inhibitor, has been shown to have anti-inflammatory and antihypertensive potential which can attenuate hypertensive-induced electrical disturbances [15]. However, the peroxisome proliferator-activated receptor-γ agonist, rosiglitazone, has been shown to increase cardiac arrhythmogenesis through dysregulated Ca^2+^ homeostasis and prolonged action in DM hypertensive rats [14]. Accordingly, we hypothesized that empagliflozin may modulate Ca^2+^ and Na^+^ homeostasis in DM cardiomyopathy, contributing to its effects on CV outcomes. Therefore, the purpose of this study was to investigate whether empagliflozin modulates cardiac electrical and structural remodeling in DM.

## 2. Results

### 2.1. Cardiac Structure and Electrocardiography of Control, DM, and Empagliflozin-Treated DM Rats

Table 1 summarizes the blood glucose, systolic and diastolic blood pressures (BPs), body weight, and heart size of the studied animals. Compared to the control rats (5.1 ± 0.1 mM, N = 7), both the DM rats (20.7 ± 2.0 mM, N = 7) and empagliflozin-treated DM rats (10.0 ± 1.3 mM, N = 7) had higher blood glucose levels (*p* < 0.005). However, the empagliflozin-treated DM rats had lower levels of blood glucose than the DM rats (*p* < 0.005). Systolic and diastolic BPs were similar in the control, DM, and empagliflozin-treated DM rats. However, the DM rats (345.5 ± 16.0 bpm) had a lower heart rate compared to the control (441.6 ± 10.8, bpm *p* < 0.005) and empagliflozin-treated rats (394.4 ± 16.0 bpm, *p* < 0.05) after treatment. Body weights were similar among the three groups before treatment, however lower body weights after treatment were noted in the DM (278.2 ± 14.9 g, *p* < 0.005) and empagliflozin-treated DM (296.2 ± 11.9 g, *p* < 0.005) groups than in the control (423.3 ± 10.3 g) group. Absolute heart weights were similar in the three groups, however the DM rats (5.4 ± 0.3 g/kg) had a greater heart-to-body weight ratio than the control, (3.2 ± 0.1 g/kg, *p* < 0.005) and than the empagliflozin-treated DM rats (4.4 ± 0.1 g/kg, *p* < 0.05).

As shown in Table 2, at 16 weeks the DM rats had higher left ventricle end-diastolic diameter (LVEDd), LV end-systolic diameter (LVESd), end-diastolic volume (EDV), end-systolic volume (ESV); and lower ejection fraction (EF) and fractional shortening (FS) values, compared to the control and empagliflozin-treated DM rats. In addition, the DM rats had longer QT intervals (90 ± 2 ms) and corrected QT intervals (QTc) (190 ± 4 ms) than either the control (QT = 70 ± 2 ms, *p* < 0.01; QTc = 170 ± 10 ms, *p* < 0.005) or empagliflozin-treated DM rats (QT = 70 ± 1 ms, *p* < 0.005; QTc = 160 ± 3 ms, *p* < 0.005, Figure 1A). However, RR intervals were similar among the three groups rat.

Moreover, measurements of the cross-sectional area in isolated single ventricular myocytes in confocal microscopic examinations showed a larger cell size in the DM group (3004 ± 81 μm^2^, *n* = 88) than in the control (2801 ± 57 μm^2^, *n* = 100, *p* < 0.05) and empagliflozin-treated DM (2635 ± 77 μm^2^, *n* = 101, *p* < 0.005) groups.

### 2.2. Effects of Empagliflozin on Action Potentials (APs) in DM Rat

DM rat ventricular myocytes had longer action potential duration (APD)_20_, APD_50_, and APD_90_ (28.4 ± 2.5 ms, 73.9 ± 6.0 ms, 198.8 ± 6.8 ms, *n* = 16) than thecontrol (13.0 ± 1.4ms, 41.6 ± 4.4 ms, 126.8 ± 8.6 ms, *n* = 16, *p* < 0.005) and empagliflozin-treated DM (18.3 ± 1.9 ms, *p* < 0.005; 63.4 ± 5.4 ms, *p* < 0.01; 151.4 ± 6.5 ms, *n* = 15, *p* < 0.005) ventricular myocytes (Figure 1B), however the control and empagliflozin-treated DM ventricular myocytes had similar APDs. Action potential amplitude (APA) and resting membrane potential (RMP) were similar in the control, DM, and empagliflozin-treated groups.

### 2.3. Effects of Empagliflozin on Ca^2+^ Stores in DM Rats

We evaluated Ca^2+^ homeostasis in the control, DM, and empagliflozin-treated DM ventricular myocytes. We found that the DM ventricular myocytes had reduced intracellular Ca^2+^ [Ca^2+^]_i_ transients (1.7 ± 0.1 F/F_0_, *n* = 45) than the control (2.3 ± 0.2 F/F_0_, *n* = 46, *p* < 0.05) and empagliflozin-treated DM (2.4 ± 0.3 F/F_0_, *n* = 46, *p* < 0.05) ventricular myocytes by 35% and 41.2%, respectively. However, the control and empagliflozin-treated DM ventricular myocytes had similar [Ca^2+^]_i_ transients (Figure 2A). In addition, the decay time (345 ± 39 ms) of [Ca^2+^]_i_ transients in the DM ventricular myocytes was significantly prolonged compared to the control (168 ± 28 ms, *p* < 0.005) and empagliflozin-treated DM (228 ± 32 ms, *p* < 0.05) ventricular myocytes, whereas the control and empagliflozin-treated DM ventricular myocytes had similar decay times of [Ca^2+^]_i_ transients. We measured the sarcoplasmic reticular (SR) Ca^2+^ content and found that the DM ventricular myocytes had significantly smaller Ca^2+^ stores (30.8 ± 4.3 μmol/L, *n* = 13) as measured by integrating the Na^+^-Ca^2+^exchanger (NCX) current after caffeine than the control (48.2 ± 8.0 μmol/L, *n* = 16, *p* < 0.05) and empagliflozin-treated DM (49.7 ± 8.0 μmol/L, *n* = 15, *p* < 0.05) ventricular myocytes (Figure 2B).

As shown in Figure 2C, the incidence and frequency of Ca^2+^ sparks were evaluated in the different groups. The incidence and frequency of Ca^2+^ sparks were increased in the DM group (9.2 ± 1.7 spark/mm/s, *n* = 76) compared to the control (1.0 ± 0.2 spark/mm/s, *n* = 40, *p* < 0.005) and empagliflozin-treated DM (1.8 ± 0.5 spark/mm/s, *n* = 36, *p* < 0.005) ventricular myocytes. In addition, the DM ventricular myocytes had a longer duration and larger width of Ca^2+^ sparks (48.0 ± 1.9 ms, 4.8 ± 0.2 μm) than the control (35.8 ± 2.7 ms, *p* < 0.005; 2.8 ± 0.2 μm, *p* < 0.005) and empagliflozin-treated DM (40.0 ± 2.1 ms, *p* < 0.05; 3.7 ± 0.2 μm, *p* < 0.005) ventricular myocytes. The peak amplitude of Ca^2+^ sparks was similar in the different groups.

### 2.4. Effects of Empagliflozin on L-type Ca^2+^ Channel (I_Ca-L_) Current and NCX Current in DM Rats

The current density of I_Ca-L_ in the DM ventricular myocytes was smaller than those in the control and empagliflozin-treated DM ventricular myocytes (Figure 3A). Figure 3B shows tracings and I-V relationships of nickel-sensitive NCX currents of ventricular myocytes from the control, DM, and empagliflozin-treated DM rats. The DM ventricular myocytes had a smaller reverse mode of nickel-sensitive NCX currents than the control and empagliflozin-treated DM ventricular myocytes.

### 2.5. Effects of Empagliflozin on Late Na^+^ Channel (I_Na-late_) Current and Na^+^/Hydrogen (Na^+^/H^+^) Exchanger Current in DM Rats

The current density of I_N**a**-late_ in the DM ventricular myocytes (0.67 ± 0.07 pA/pF, *n* = 15) was greater than those in the control (0.49 ± 0.04 pA/pF, *n* = 18, *p* < 0.05) and empagliflozin-treated DM (0.51 ± 0.03 pA/pF, *n* = 16, *p* < 0.05) ventricular myocytes (Figure 4A). Figure 4B shows tracings and current densities of Na^+^/H^+^ exchanger currents of ventricular myocytes from the control, DM, and empagliflozin-treated DM rats.

The DM ventricular myocytes (17.1 ± 1.6 pA/pF, *n* = 18) had a larger Na^+^/H^+^ exchanger current than the control (11.6 ± 1.3pA/pF, *n* = 17, *p* < 0.05) and empagliflozin-treated DM (12.1 ± 1.1pA/pF, *n* = 19, *p* < 0.05) ventricular myocytes.

### 2.6. Effects of Empagliflozin on Oxidative Stress in DM Rats

As shown in Figure 5A, the DM ventricular myocytes (126 ± 5 F/F_0_, *n* = 28) had higher levels of cytosolic reactive oxygen species (ROS) than the control (89 ± 10 F/F_0_, *n* = 21, *p* < 0.005) and empagliflozin-treated DM (88 ± 5 F/F_0_, *n* = 46, *p* < 0.005) ventricular myocytes, by 28.8% and 29.8%, respectively. The control and empagliflozin-treated DM ventricular myocytes had similar cytosolic ROS levels. In addition, the level of mitochondrial ROS in the DM ventricular myocytes (41 ± 1 F/F_0_, *n* = 38) was greater than those in the control (23 ± 1 F/F_0_, *n* = 50, *p* < 0.005) and empagliflozin-treated DM (37 ± 2 F/F_0_, *n* = 28, *p* < 0.05) ventricular myocytes by 44.3% and 10.5%, respectively (Figure 5B). Similarly, the intracellular Na^+^ ([Na^+^]_i_) level in the DM ventricular myocytes (133 ± 18 F/F_0_, *n* = 22) was greater than those in the control (115 ± 22 F/F_0_, *n* = 29, *p* < 0.005) and empagliflozin-treated DM (122 ± 17 F/F_0_, *n* = 27, *p* < 0.05) ventricular myocytes by 13.2% and 7.8%, respectively (Figure 5C).

### 2.7. Acute Effects of Empagliflozin on DM Rat Ventricular Myocytes

In order to study whether empagliflozin may directly affect cardiac disorder in DM rats, we investigated the acute effects of empagliflozin (1 μM) on isolated DM rat cardiomyocytes. As shown in Figure 6, acute administration of empagliflozin (1 μM) reduced the APD_90_ (from 170 ± 4 ms to 129 ± 11 ms, *n* = 10, *p* < 0.01), and APD_50_ (from 77 ± 6 ms to 60 ± 6 ms, *n* = 10, *p* < 0.05) in DM ventricular myocytes. In addition, empagliflozin (1 μM) reduced I_Na-Late_ from 0.66 ± 0.07 pA/pF to 0.28 ± 0.04 pA/pF (*n* = 11, *p* < 0.005) and cytosolic ROS levels from 108 ± 4 F/F_0_ to 81 ± 4 F/F_0_ (*n* = 11, *p* < 0.005) in DM ventricular myocytes.

### 2.8. Effects of Empagliflozin on Ca^2+^ Regulatory Proteins in DM Rats

We compared the expressions of Ca^2+^ regulatory proteins in control, DM and empagliflozin-treated DM rat ventricles using Western blotting (Figure 7). The protein expressions of SERCA2a (0.57 ± 0.06, relative to control, *p* < 0.005) and RyR2 (0.69 ± 0.09 relative to control, *p* < 0.05) in the DM group (N = 7) were downregulated, compared with those in control (N = 7) and empagliflozin-treated DM groups (0.98 ± 0.09; 1.09 ± 0.12, relative to control, N = 7, Figure 7). In contrast, the protein level of RyR2-pS2808 in the DM group (1.51 ± 0.17, relative to control, N = 7) was higher than those in the control (N = 7, *p* < 0.05 and empagliflozin-treated DM groups (0.53 ± 0.04, relative to control, *p* < 0.005; N = 7).

## 3. Discussion

Altered expressions, activities and functions of transporters involved in excitation-contraction coupling, i.e., SERCA [17], NCX [18], and RyR [19], as well as dysfunction in [Ca^2+^]_i_ signaling [20] have been reported in DM rodent models. However, in the current investigation, we found that myocardial dysfunction, prolonged ventricular APs, decreased Ca^2+^ transients, increased I_N**a**-late_ and Na^+^/H^+^-exchanger currents in the DM rats were attenuated after the administration of empagliflozin. Therefore, empagliflozin may affect the electromechanical mechanisms in DM cardiomyopathy.

DM patients have a high incidence of DM cardiomyopathy, characterized by complex changes in the mechanical and electrical properties of the heart [21]. DM patients usually present with prolongation of both QT and QTc intervals due to an increase in ventricular APD that makes them more susceptible to an increased incidence of arrhythmogenesis [22,23]. In this study, we found that the DM rats had a greater heart-to-body weight ratio and larger cross-sectional area of ventricular myocytes than the control and empagliflozin-treated groups. These findings are compatible with the results of a previous study [24]. The DM rats had prolonged QT and QTc intervals and ventricular APD, as we also demonstrated in our previous study [14]. Prolongation of the APD exacerbates the decrease in diastolic filling and stroke volume at high heart rates [25], and prolonged QT intervals may increase ventricular arrhythmia due to triggered activity of early afterdepolarization [14]. Moreover, we also found that empagliflozin changed the AP morphology and cardiac electrophysiology of the DM hearts. Reversed prolongation of the APD in the ventricular myocytes of DM rats after treatment suggests that empagliflozin may have a cardioprotective effect during electrophysiological alterations of DM hearts.

Lambert et al. found a higher [Na^+^]_i_ in late-onset type-2 DM rat myocytes which displayed a cardiac phenotype that closely resembled human DM cardiomyopathy [12]. The increase in I_N**a**-late_ and Na^+^/H^+^-exchanger activity may contribute to the elevation of [Na^+^]_i_ of DM myocytes, leading to Ca^2+^ overload, with an increased risk of arrhythmias and oxidative stress [26]. Similar to previous studies, our results showed that the ventricular myocytes of the DM rats had a reduced [Ca^2+^]_i_ transient amplitude, prolonged transient decay, and reduced Ca^2+^ store [14,27,28]. Ca^2+^ is one of the main ionic regulators of the heart, and it plays an important role in the process of excitation-contraction coupling.

Ca^2+^ activates the release of further ionic Ca^2+^ from SR stores through RyR, which increases [Ca^2+^]_i_ and aids in the binding of Ca^2+^ to myofilaments and initiating cardiac contractions [29]. The decrease in [Ca^2+^]_i_ transients in our DM hearts may have been caused by a decrease in SR Ca^2+^ content, resulting in impaired excitation-contraction coupling efficiency and myocardial dysfunction. The reduction in SR Ca^2+^ stores might have been exacerbated in DM hearts due to a decrease in SERCA2a protein. We also found a prolonged [Ca^2+^]_i_ transient decay associated with the depletion of SR Ca^2+^ contents in the DM hearts. The increase in the [Ca^2+^]_i_ transient decay may have been due to derangement in either the SERCA pump resulting in a decrease in the removal rate of cytoplasmic Ca^2+^ [18,27], and this may have caused impaired relaxation of the cardiomyocytes. The slower kinetics of the [Ca^2+^]_i_ transient of DM ventricular myocytes may also have been affected by the prolonged APD, altered [Ca^2+^]_i_ handling, or a combination of both. We also found that decreases in [Ca^2+^]_i_ transients and Ca^2+^ stores with prolonged [Ca^2+^]_i_ decay in the DM hearts were attenuated in the DM rats treated with empagliflozin. These effects may have been caused by enhanced myocardial SERCA2a function as demonstrated in a previous study by Hammoudi et al. [30] resulting in increased reuptake of [Ca^2+^]_i_ after [Ca^2+^]_i_ release in the DM rats treated with empagliflozin. In addition, the Ca^2+^ regulatory effects of empagliflozin in DM hearts may have contributed to the improved cardiac function measured by echocardiography. The increase in SERCA2a protein in the DM rats treated with empagliflozin attenuated the negative impact of DM on Ca^2+^ dysregulation. These findings suggest that empagliflozin has potential to be a therapeutic strategy for cardiac electrical dysfunction in DM.

DM has been shown to modify Ca^2+^ entry through molecular mediators that permit the Ca^2+^ influx required for excitation-contraction coupling [31]. Several studies have shown that Ca^2+^ entry through voltage-dependent I_Ca-L_ is reduced in the cardiac myocytes of DM rodents [32,33,34]. I_Ca-L_ is a critical initiator of the contractile cycle in cardiac myocytes, and inhibition of I_Ca-L_ function has been reported to potentially reduce the entry of Ca^2+^ resulting in decreases in Ca^2+^ transients and contractile force [35]. This may also have caused the prolongation of QT and QTc intervals in electro-cardiography. Moreover, reversal of the I_Ca-L_ in the DM rats treated with empagliflozin may have contributed, at least in part, to the improved [Ca^2+^]_i_ transients and SR Ca^2+^ contents. In this study, the reverse mode (Ca^2+^ influx) of NCX was depressed in the DM ventricular myocytes, indicating that a reduction in Ca^2+^ influx may have contributed to the decrease in [Ca^2+^]_i_. We also found that empagliflozin treatment restored the diminished NCX current.

Ca^2+^ sparks are brief releases of Ca^2+^ that are considered to be essential events in the excitation-contraction of cardiomyocyten addition, the frequency and incidence of Ca^2+^ sparks in the DM rats were also reversed after empagliflozin treatment. Thus, empagliflozin may have a beneficial effect in preventing increased Ca^2+^ leakage in the SR that may cause Ca^2+^ deficiency leading to myocardial dysfunction in DM cardiomyopathy. The decrease in RyR phosphorylation in DM hearts receiving empagliflozin may increase the amplitude of [Ca^2+^]_i_ transients and improve myocardial contractility.

We evaluated the chronic in vivo effect of empagliflozin treatment on I_Ca-L_ and Na^+^-H^+^ exchanger currents in DM ventricular myocytes. We demonstrated that chronic treatment with empagliflozin lowered intracellular Na^+^ levels, and that I_Na-late_ and Na^+^-H^+^ exchange currents were ameliorated after empagliflozin treatment. This may at least in part be related to the effect of empagliflozin on cardiac ion homeostasis, and may contribute to the cardiac benefits in DM patients. Baartscheer et al. [36] have found that empagliflozin (1 μM) directly regulates cytoplasmic Na^+^ and Ca^2+^ in cardiomyocytes. Our study also found that acute administration of empagliflozin (1 μM) reduced APD, I_Na-late_, and ROS in DM ventricular myocytes, which was similar to the results seen from the in vivo treatment of empagliflozin in DM rat model. Previous study has shown that patients have peak plasma concentration of empagliflozin close to 1 μM [37], thus the concentration of empagliflozin used in this study is clinically relevant. These findings suggest that empagliflozin may have direct effects on cardiac disorder in DM rats. However, the hypoglycemic effects of empagliflozin may have also contributed to the improvements in the cardiac condition in DM cardiomyopathy.

We evaluated ROS production in the DM hearts, and found that both cytosolic and mitochondria ROS production were decreased after treatment with empagliflozin. Ca^2+^ sparks induced by RyR hyperphosphorylation can lead to mitochondrial Ca^2+^ overload, thereby facilitating ROS production.

In addition, mitochondria-derived ROS can induce local ER Ca^2+^ release events, which increased spark frequency in cardiomyocytes. The reduction of Ca^2+^ sparks by empagliflozin may contribute to its inhibitory effect on ROS production [38]. Moreover, the hypoglycemic effects of empagliflozin may reduce ROS, since hyperglycemia and insulin resistance result in excess ROS production [39]. The reduction in ROS production after empagliflozin treatment may have contributed to attenuating the increase in Ca^2+^sparks in the empagliflozin-treated DM hearts. However, characterization of the mechanical function in arrhythmogenesis and heart failure models are needed to further understand the cardioprotective effect of empagliflozin in DM hearts. We also found that the blood glucose level was lower in our STZ-induced DM rats treated with empagliflozin. Improvements in glycemic control through non-insulin-dependent pathways with SGLT2 inhibitors are also important as an add-on treatment for type 1 DM, as improved glycemic control can reduce end-organ complications in DM patients. The lower I_N**a**-late_, Na^+^/H^+^ exchanger, ROS and Ca^2+^ sparks in the empagliflozin-treated DM ventricular myocytes than in the DM ventricular myocytes suggests that empagliflozin may possess anti-arrhythmic potential. Therefore, it would be interesting to perform further tests to verify the anti-arrhythmic effect of empagliflozin. Figure 8 details the potential action mechanisms of empagliflozin in the DM hearts in this study. Empagliflozin treatment may reverse DM-induced Ca^2+^/Na^+^ dysregulation by decreasing ROS and ionic channel modification in cardiomyocytes, leading to improvements in cardiac function, attenuation of ventricular hypertrophy, and correction of prolonged QT intervals.

### Study Limitation

First, T-type Ca^2+^ channels have been shown to be re-expressed in hypertrophic ventricular myocytes [40]. However, we did not measure T-type Ca^2+^ channels in this study because they contribute less to the trigger for Ca^2+^ release [40]. Second, APD is controlled not only by Ca^2+^ or Na^+^ channels but also by potassium (K^+^) channels.

However, we mainly focused on the effects of empagliflozin on Ca^2+^/Na^+^ regulation in DM cardiomyocytes, and did not examine the functional changes in K^+^ channels. Finally, the experimental setting in this study may not fully correlate with clinical features. Most DM patients have faster resting heart rate due to autonomic neuropathy [41,42]. On the contrary, the DM rats had a lower heart rate compared to the control in this study. STZ-induced DM resembles human type 1 DM, which is associated with a lower body weight. Nevertheless, most DM patients have a higher body weight than subjects without DM, since type 2 DM is more common. This study showed that empagliflozin treatment did not reverse the lower body weight in DM rats, which may have been caused by inadequate glycemic control.

## 4. Methods

### 4.1. Ethical Approval

This investigation conformed to the institutional Guide for the Care and Use of Laboratory Animals and the Guide for the Care and Use of Laboratory Animals published by the US National Institutes of Health (NIH publication no. 85-23, revised 1996), and was approved on April 25, 2016 by the Institutional Animal Care and Use Committee of Taipei Medical University (LAC-2016-0425).

### 4.2. Induction of DM, Treatment and Tissue Harvesting

Rats were housed in standard environmental conditions of low humidity and temperature (21 ± 2°C) with a 12-h light/dark cycle, and were maintained on commercial rat chow and tap water, ad libitum. To induce diabetes mellitus (DM), some of the 10-week-old male Wistar rats (~335 ± 4.5 g) received a single intraperitoneal injection of STZ (65 mg/kg, Sigma-Aldrich; St. Louis, MO, USA) after 10 h of overnight starvation [14,43]. DM was diagnosed according to a high fasting plasma glucose (≥15 mmol/L) [14,43], as measured with a glucometer (Bayer Breeze 2 glucometer, Bayer Health Care, Mishawaka, IN, USA). Two weeks after the induction of DM (12 weeks of age), DM rats were randomly assigned to receive empagliflozin (10 mg/kg, po, Jardiance, Boehringer Ingelheim Pharmaceuticals, Inc., Ridgefield, CT, USA) [44] or vehicle (1 cc normal saline) once daily for 4 weeks by oral gavage. The rats were anesthetized by deep inhalation with 5% isoflurane [45], and sacrificed at 16 weeks of age. Body weights were measured prior to euthanasia. Each heart was rapidly excised, weighed, and dissected. Freshly isolated ventricular tissues were rinsed in a cold physiological saline solution and frozen in liquid nitrogen for protein isolation.

### 4.3. Echocardiography and Electrocardiography Measurements

Under isoflurane anesthesia (5% for induction and 2% for maintenance), transthoracic echocardiography was performed using a Vivid I ultrasound cardiovascular system (GE Healthcare, Haifa, Israel) at 16 weeks of age (6 weeks after receiving STZ) in the control and DM rats, with and without empagliflozin treatment. The following cardiac structures were measured using M-Mode tracing of the LV: LVEDd, LVESd, EDV, ESV, FS, and EF [46].

Electrocardiography was performed at 10 and 16 weeks of age, and electrocardiograms were recorded from standard lead II limb leads via a bio-amplifier (AD Instruments, Castle Hill, Australia), connected to a polygraph recorder (ML 845 Powerlab, AD Instruments) [47]. The results were continuously displayed throughout the experiments in the control and DM rats, with and without empagliflozin treatment.

The systolic and diastolic BPs of the control and DM rats with and without empagliflozin treatment, were measured at 10 and 16 weeks of age using a non-invasive BP tail-cuff method (MK-2000, Muromachi Kikai, Tokyo, Japan) as described previously [47].

### 4.4. Measurement of [Ca^2+^]_i_, SR Ca^2+^ Contents, and Ca^2+^ Spark Imaging

Freshly isolated rat ventricular myocytes were enzymatically dissociated using collagenase (type I, Sigma) and protease (type XIV, Sigma) after euthanasia as described previously [14]. [Ca^2+^]_i_ was recorded in single myocytes. Cells were loaded with a fluorescent Ca^2+^ (10 μM) fluo-3/AM for 30 min at room temperature. Excess extracellular dye was removed by changing the bath solution and allowing the intracellular hydrolysis of fluo-3/AM after 30 min. Fluo-3 fluorescence was excited with a 488-nm line of an argon ion laser. The emission was recorded at >515 nm. For line scan imaging (8-bit), cells were repetitively scanned at 2-ms intervals. Fluorescence imaging was performed using a laser scanning confocal microscopic examination (Zeiss LSM 510, Carl Zeiss, Jena, Germany) and an inverted microscope (Axiovert 100, Carl Zeiss, Jena, Germany). The fluorescent signals were corrected for variation in dye concentration by normalizing the fluorescence (F) against baseline fluorescence (F_0_), to obtain reliable information about transient intracellular Ca^2+^ ([Ca^2+^]_i_) changes from the baseline values (F/F_0_) and to exclude variations in the fluorescence intensity by different volumes of injected dye. The [Ca^2+^]_i_ transient, and decayed portion of the [Ca^2+^]_i_ transient were measured during a 1-Hz field-stimulation with 10-ms twice-threshold strength square-wave pulses. The [Ca^2+^]_i_ was determined by the monoexponential least-squares fit. After achieving steady-state Ca^2+^ transients with repeated pulses from -40 to 0 mV (1 Hz for 5 s), the SR Ca^2+^ content was estimated by integrating the NCX current following application of 20 mM of caffeine within 0.5 s while resting with the membrane potential clamped to −40 mV to cause SR Ca^2+^ release [48]. The total SR Ca^2+^ content (expressed as millimoles of cytosol) was determined using this equation: SR Ca^2+^ content = [(1 + 0.12) (Ccaff/F × 1000)]/(Cm × 8.31 × 8.44), where Cm is the membrane capacitance; F is Faraday’s number; and the cell surface to volume ratio is 6.44 pF/pL [49,50].

Ca^2+^ sparks were detected using the line scan mode along a line parallel to the longitudinal axis of single freshly isolated rat ventricular myocytes, while avoiding the nuclei. Each line was composed of 512 pixels. Ca^2+^ sparks were detected as an increase in the signal mass (more than 1.3 *F*/*F*_0_) of a 5-µm section through the center of a Ca^2+^ spark, with no detectable increase in the adjacent 5-µm section. The Ca^2+^ spark frequency was determined for each cell and normalized to the scanned cell length. Images were analyzed using both ImageJ and custom-made routines based on IDL (ITT Visual Information Solutions, CO, USA).

### 4.5. AP and Ionic Currents

A whole-cell patch-clamp using an Axopatch 1D amplifier (Axon Instruments, Foster City, CA, USA) was performed in the freshly isolated ventricular myocytes of rats at 35 ± 1 °C. Borosilicate glass electrodes (o.d., 1.8 mm) were used with tip resistances of 3~5 MΩ. Before the formation of the membrane-pipette seal, the tip potentials were zeroed in Tyrode’s solution. Junction potentials between the bath and pipette solution (9 mV) were corrected for AP recordings. APs driven at 1 Hz were recorded in the current-clamp mode, and ionic currents were measured in the voltage-clamp mode. A small hyperpolarizing step from a holding potential of −50 mV to a test potential of −55 mV for 80 ms was delivered at the beginning of each experiment. The area under the capacitative currents curve was divided by the applied voltage step to obtain the total cell capacitance [51]. In most cases, 60%~80% series resistance was electronically compensated. Micropipettes were filled with solutions containing (in mM): CsCl 130, MgCl_2_ 1, MgATP 5, HEPES 10, EGTA 10, NaGTP 0.1, and Na_2_ phosphocreatine 5, titrated to a pH of 7.2 with CsOH for I_Ca-L_; NaCl 20, CsCl 110, MgCl_2_ 0.4, CaCl_2_ 1.75, tetraethylammonium 20, 1,2-bis(2-aminophenoxy)ethane-N,N,N’,N’-tetraacetic acid (BAPTA) 5, glucose 5, MgATP 5, and HEPES 10, titrated to pH of 7.25 for NCX currents; 10 NaCl, 130 CsCl, 5 EGTA, 5 HEPES, 5 glucose, and 5 ATP-Mg for the I_Na-Late_; and KCl 20, K aspartate 110, MgCl_2_ 1, MgATP 5, HEPES 10, EGTA 0.5, LiGTP 0.1, and Na_2_ phosphocreatine 5, titrated to a pH of 7.2 with KOH for the APs. Voltage command pulses were generated with a 12-bit digital-to-analog converter controlled using pCLAMP software (Axon Instruments). Recordings were low pass-filtered at half the sampling frequency. The APA was determined by measuring the difference between the RMP and the peak of AP depolarization. The APD at repolarization extents of 90%, 50%, and 20% of the amplitude were measured, and respectively designated as APD_90_, APD_50_, and APD_20_.

I_Ca-L_ was measured as an inward current during depolarization from a holding potential of −50 mV to test potentials ranging from −40 to +60 mV in 10-mV steps for 300 ms at a frequency of 0.1 Hz. NaCl and KCl in the external solution were respectively replaced by tetraethylammonium chloride and CsCl. The NCX current was elicited by depolarizing pulses between −100 and +100 mV from a holding potential of −40 mV for 300 ms at a frequency of 0.1 Hz. Amplitudes of NCX currents were measured as 10-mM nickel-sensitive currents. The external solution consisted of (in mM) NaCl 140, CaCl_2_ 2, MgCl_2_ 1, HEPES 5, and glucose 10, with the pH adjusted to 7.4 and 10 μM strophanthidin (to block the Na^+^/K^+^ pump), 10 μM nitrendipine (a dihydropyridine antagonist), and 100 μM niflumic acid (to block Ca^2+^-activated Cl^−^ currents).

I_Na-Late_ was recorded with an external solution containing (in mM): NaCl 130, CsCl 5, MgCl_2_ 1, CaCl_2_ 1, HEPES 10, and glucose 10 at a pH of 7.4 with NaOH by a step/ramp protocol (−100 mV stepped to +20 mV at room temperature for 100 ms, then ramped back to −100 mV over 100 ms). The tetrodotoxin (30 μM)-sensitive portion of the current traces, obtained when the voltage was ramped back to −100 mV, was measured as the I_Na-Late_ current [51].

To measure Na^+^/H^+^ exchanger current, a whole-cell camp with patch electrodes of 1.5–3.0 MΩ resistance were filled with solution consisting of (in mM) 20 KCl, 130 K-Aspartate, 1 MgCl_2_ (6 H_2_O), 10 HEPES/KOH (pH 7.3), and 0.005 EGTA. The external solution consisted of (in mM) 150 NaCl, 5.4 KCl, 3.6 CaCl_2_, 1.2 MgCl_2_ (6 H_2_O), 20 glucose, and 5 HEPES/NaOH (pH 7.4). All groups of freshly isolated rat ventricular myocytes were stimulated with 40 ms depolarizing pulses from −45 to 0 mV at 3 Hz [52].

### 4.6. Measurement of Cell Size and Intracellular ROS and Na^+^

The cross-sectional area in the isolated single ventricular myocytes were imaged using a confocal laser scan microscope (Zeiss LSM 510, Carl Zeiss) and images were processed by ImageJ measurement tools [53]. We used CellROX green (Life Technologies, Grand Island, NY, USA) to assess cytosolic ROS production, MitoSOX Red (Life Technologies) to determine ROS production in the mitochondria, and Asante NaTRIUM Green-2 AM (Teflabs, Austin, TX, USA) to evaluate the cytosolic Na^+^ concentration of freshly isolated ventricular myocytes in the control, DM, and empagliflozin-treated DM rats. Experiments were carried out using a laser scanning confocal microscope (Zeiss LSM 510, Carl Zeiss) and an inverted microscope (Axiovert 100) with a 63x1.25 numerical aperture oil immersion objective, as described previously [54]. Freshly isolated ventricular myocytes were kept in normal Tyrode’s solution (containing in mM): NaCl 137, KCl 5.4, CaCl_2_ 1.8, MgCl_2_ 0.5, and HEPES 10, with appropriate fluorescent dye of 10 μM CellROX green, 2 μM MitoSOX Red, or 5 μmol/L Asante NaTRIUM Green-2 AM. CellROX green, MitoSOX Red, and Asante NaTRIUM Green-2 AM were excited at 488 nm, and fluorescence signals were acquired at wavelengths of >505 nm in the XY mode of the confocal system. In this experiment, ventricular myocytes were paced at 1 Hz. Fluorescent images were analyzed using Image-Pro plus 6.0 and Sigma plot 12 software, as described previously [51].

### 4.7. Western Blot Analysis

Equal amounts of proteins were resolved using sodium dodecyl sulfate polyacrylamide gel electrophoresis as described previously [43]. Blots were probed with antibodies against sarcoplasmic reticulum ATPase (SERCA2a, cat. no. Sc-8094; 1:1000; Santa Cruz Biotechnology, CA, USA), ryanodine receptor 2 (RyR2; cat. no. MA3-916; 1:1000; Affinity Bioreagents, Golden, CO, USA), phosphorylated RyR2 at serine 2808 (RyR2-pS2808; cat. no. A010-30AP; 1:000; Badrilla, UK), and secondary antibodies conjugated with horseradish peroxidase (Leinco Technology, St. Louis, MO, USA). Bound antibodies were detected using an enhanced chemiluminescence detection system (Millipore, St. Louis, MO, USA) and analyzed with AlphaEaseFC software (Alpha Innotech, San Leandro, CA, USA).

Targeted bands were normalized to cardiac glyceraldehyde 3-phosphate dehydrogenase (GAPDH; cat. no. M171-7; 1:1000; Sigma-Aldrich) to confirm equal protein loading.

### 4.8. Statistical Analysis

All quantitative data are expressed as the mean ± standard error of the mean (SEM). Statistical significance between different groups was determined using an unpaired *t*-test or one-way analysis of variance (ANOVA) with Duncan’s method, using Systat software SigmaPlot version 12 (Systat Software Inc., San Jose, CA, USA) for multiple comparisons as appropriate. A *p* value of <0.05 was considered to be statistically significant. The sample size was calculated according to the ejection fraction with an estimated standard deviation of 3. The required sample size was 7 under a type I error of 5% with a statistical power of 80%.

## Figures and Tables

**Figure 1 ijms-20-01680-f001:**
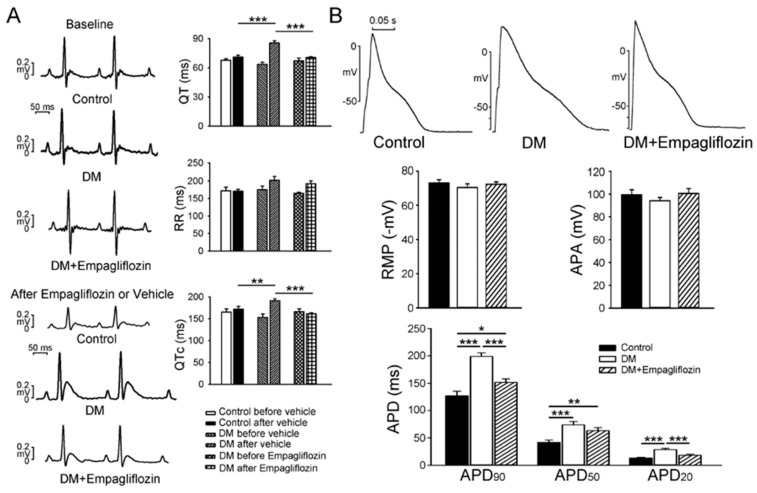
Electrocardiographic changes and action potentials (APs) of ventricular myocytes in control, diabetes mellitus (DM), and empagliflozin-treated DM (DM + empagliflozin) rats. (**A**) Representative electrocardiographic tracings and average data before and after treatment in control (N = 7), DM (N = 7), and DM + empagliflozin (N = 7) rats. (**B**) Representative tracings and average data of APs in control (*n* = 16), DM (*n* = 16), and DM + empagliflozin ventricular myocytes (*n* = 15). QT: QT interval; RR: RR interval; QTc: Corrected QT interval; N = number of rats; *n* = number of cardiomyocytes isolated from those rats; RMP: Resting Membrane Potential; APA: Action Potential Amplitude; APD_20_, APD_50_, and APD_90_: Action potential durations at 20%, 50%, and 90% of repolarization, respectively; * *p* < 0.05; ** *p* < 0.01; *** *p* < 0.005.

**Figure 2 ijms-20-01680-f002:**
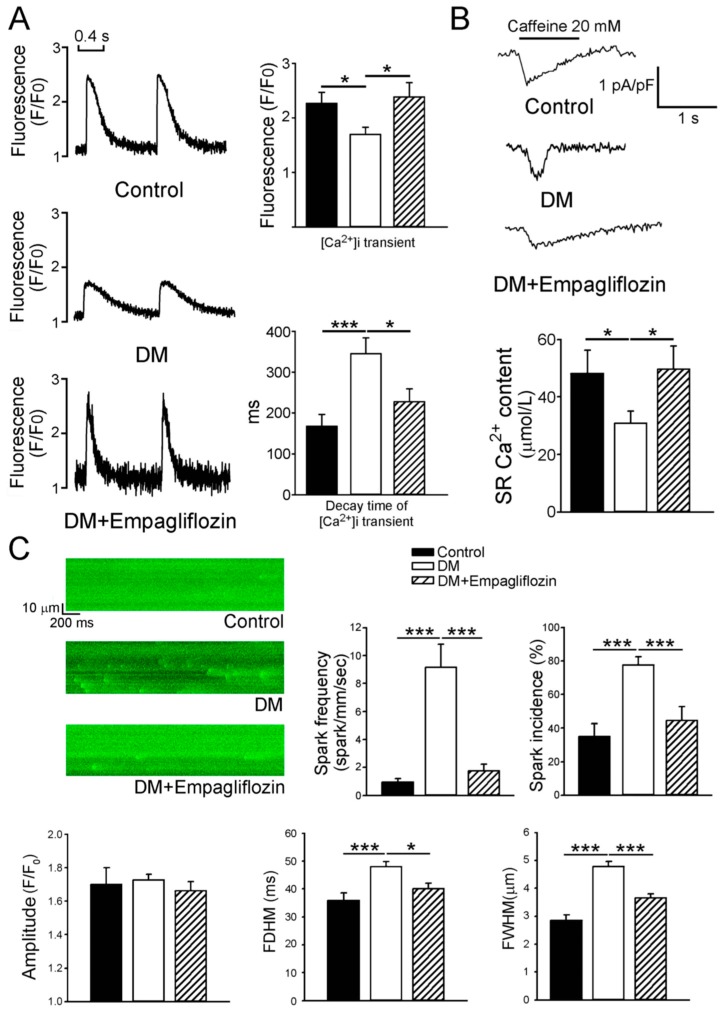
Intracellular Ca^2+^ ([Ca^2+^]_i_) transients, calcium (Ca^2+^) stores measured from caffeine (20 mM)-induced Ca^2+^ transients and Ca^2+^ sparks in ventricular myocytes from control, diabetes mellitus (DM), and empagliflozin-treated DM (DM + empagliflozin) rats. (**A**) Representative tracings and average data of [Ca^2+^]_i_ transients and decay of time of [Ca^2+^]_i_ transients in the ventricular myocytes recorded from control (*n* = 46), DM (*n* = 45), and DM + empagliflozin (*n* = 46) rats. (**B**) Representative tracings and average data of Ca^2+^ stores of ventricular myocytes recorded from the control (*n* = 16), DM (*n* = 13), and DM + empagliflozin (*n* = 15) rats. (**C**) Representative tracings and average data of the incidence, frequency, amplitude, duration (full duration at half-maximal amplitude, FDHM), and width (full-width at half-maximal amplitude, FWHM) of Ca^2+^ sparks in control (*n* = 40), DM (*n* = 76), and DM + empagliflozin (*n* = 36) ventricular myocytes. *n* = number of cardiomyocytes isolated from those rats; * *p* < 0.05; *** *p* < 0.005.

**Figure 3 ijms-20-01680-f003:**
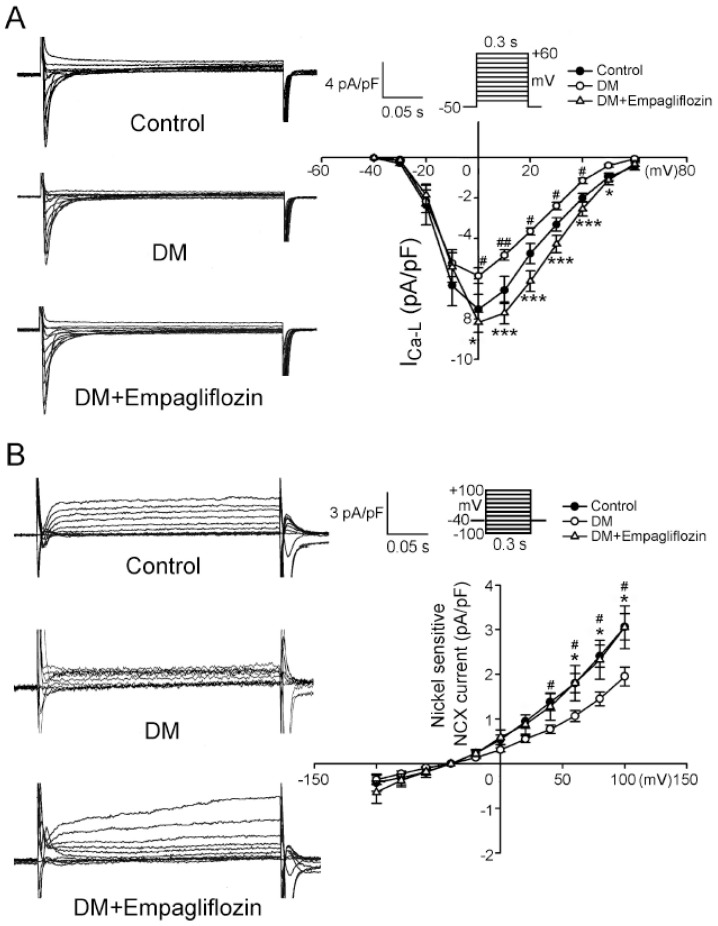
L-type calcium channel (I_Ca-L_) and Na^+^-Ca^2+^ exchanger (NCX) current of ventricular myocytes from control, diabetes mellitus (DM), and empagliflozin-treated DM (DM + empagliflozin) rats. (**A**) Representative tracings of current and I-V relationship of the I_Ca-L_ of ventricular myocytes from control (*n* = 11), DM (*n* = 14), and DM + empagliflozin (*n* = 21) rats. (**B**) Representative tracings of current and I-V relationship of the NCX current of ventricular myocytes from control (*n* = 11), DM (*n* = 13), and DM + empagliflozin (*n* = 8) rats. *n* = number of cardiomyocytes isolated from those rats; ^#^
*p* < 0.05 vs. the controls; ^##^
*p* < 0.01 vs. the controls; * *p* < 0.05 vs. DM + empagliflozin rats; *** *p* < 0.005 vs. DM + empagliflozin rats.

**Figure 4 ijms-20-01680-f004:**
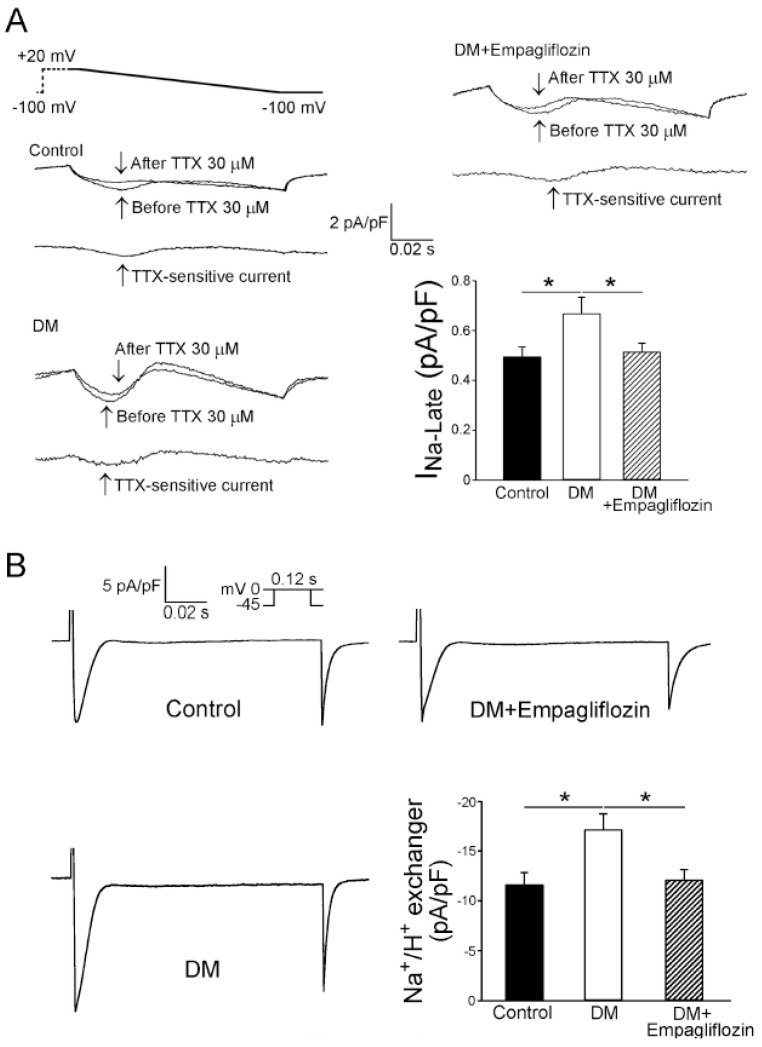
Late sodium current (I_Na-Late_) and sodium hydrogen (Na^+^/H^+^) exchanger current of ventricular myocytes from control, diabetes mellitus (DM), and empagliflozin-treated DM (DM + empagliflozin) rats. (**A**) Representative tracing of current and average data of the I_Na-Late_ of ventricular myocytes from control (*n* = 18), DM (*n* = 15), and DM + empagliflozin (*n* = 16) rats. (**B**) Representative tracings of current and average data of the Na^+^/H^+^ exchanger of ventricular myocytes from control (*n* = 17), DM (*n* = 18), and DM + empagliflozin (*n* = 19) rats. *n* = number of cardiomyocytes isolated from those rats; * *p* < 0.05.

**Figure 5 ijms-20-01680-f005:**
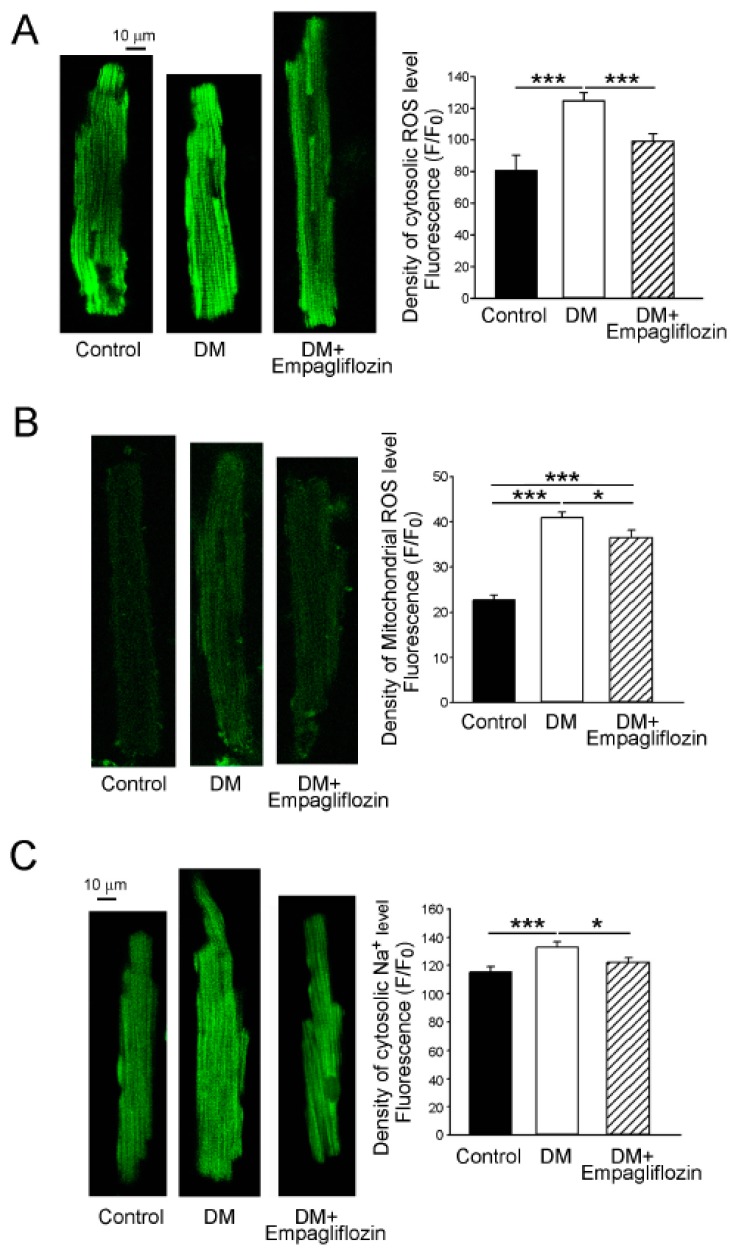
Oxidative stress and cytosolic sodium (Na^+^) levels in control, diabetes mellitus (DM), and empagliflozin-treated DM (DM + empagliflozin) ventricular myocytes. (**A**) An example and average data of cytosolic levels of reactive oxygen species (ROS) in control (*n* = 21), DM (*n* = 28), and DM + empagliflozin (*n* = 46) ventricular myocytes. (**B**) An example and average data of mitochondrial levels of ROS in control (*n* = 50), DM (*n* = 38), and DM + empagliflozin (*n* = 28) ventricular myocytes. (**C**) An example and average data of cytosolic Na^+^ levels in control (*n* = 29), DM (*n* = 22), and DM + empagliflozin (*n* = 27) ventricular myocytes. *n* = number of cardiomyocytes isolated from those rats; * *p* < 0.05; *** *p* < 0.005.

**Figure 6 ijms-20-01680-f006:**
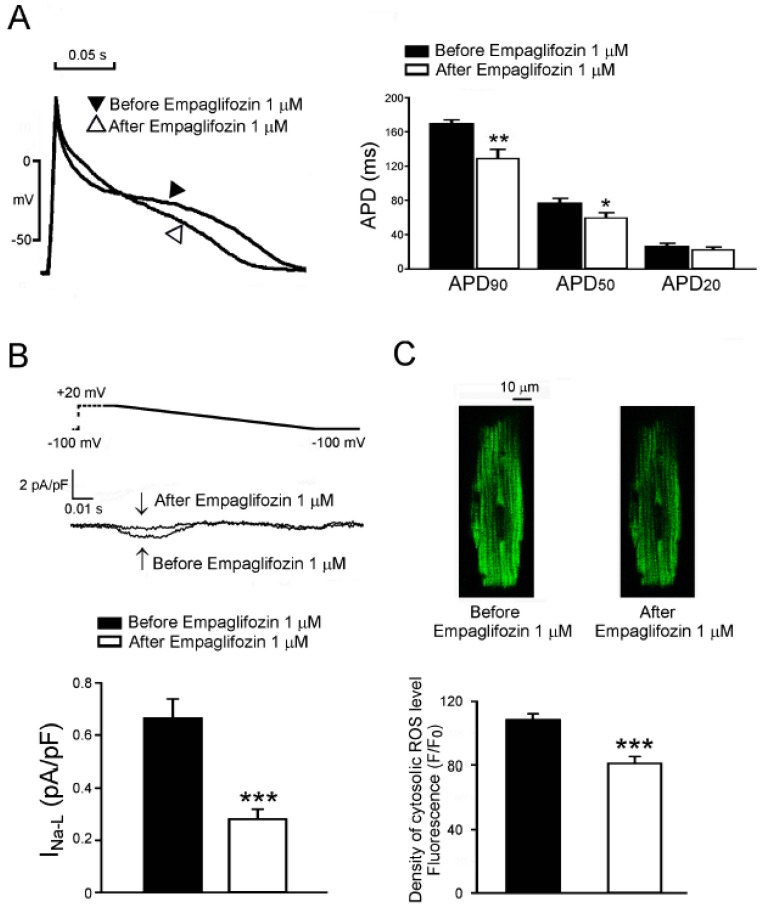
Acute effects of empagliflozin on DM ventricular myocytes. (**A**) The superimposed tracings and average data (*n* = 10) of AP morphology before and after acute administration of empagliflozin (1 μM) in DM rat ventricular myocytes. (**B**) Representative superimposed tracing of current and average data (*n* = 11) of the I_Na__-Late_ before and after acute administration of empagliflozin (1 μM) in DM rat ventricular myocytes. (**C**) An example and average data (*n* = 11) of cytosolic levels of reactive oxygen species (ROS) before and after acute administration of empagliflozin (1 μM) in DM rat ventricular myocytes. * *p* < 0.05; ** *p* < 0.01; *** *p* < 0.005 versus before empagliflozin.

**Figure 7 ijms-20-01680-f007:**
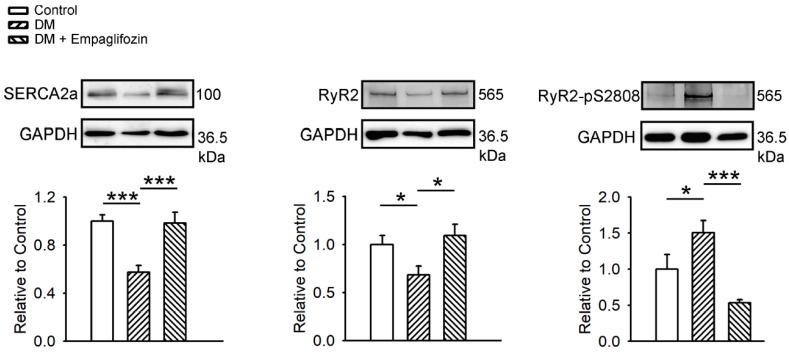
Ca^2+^ regulation proteins in control, diabetes mellitus (DM), and empagliflozin-treated DM (DM + empagliflozin) ventricular myocytes. Representative immunoblot and average data of sarcoplasmic reticulum ATPase (SERCA2a), ryanodine receptor 2 (RyR2), and phosphorylated RyR2 at serine 2808 (RyR2-pS2808) from control (N = 5), DM (N = 5), and DM + empagliflozin (N = 5) rat ventricular myocytes. Densitometry was normalized to glyceraldehyde 3-phosphate dehydrogenase (GAPDH) as an internal control. N = number of rats; * *p* < 0.05; *** *p* < 0.005.

**Figure 8 ijms-20-01680-f008:**
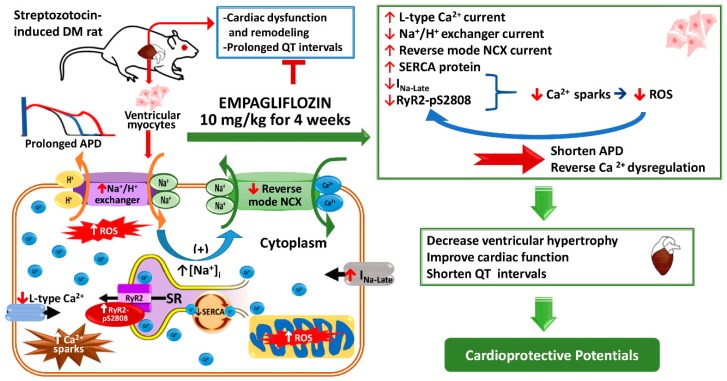
Schematic illustration of the proposed mechanism of action of empagliflozin in DM hearts. Empagliflozin treatment may reverse DM-induced Ca^2+^/Na^+^ dysregulation by decreasing ROS and ionic channel modification in cardiomyocytes, leading to improvements in cardiac function, attenuation of ventricular hypertrophy, and correction of prolonged QT intervals. APD: Action Potential Duration DM: Diabetes Mellitus; Ca^2+^: Calcium; H^+^: Hydrogen; [Na^+^]_i_: Intracellular Sodium; I_Na__-Late_: Late Sodium Current; Na^+^: Sodium; NCX: Sodium Calcium Exchanger; QT: QT interval; Na^+^: Sodium; ROS: Reactive Oxygen Species; RyR2: Ryanodine Receptor; RyR2-pS2808: Phosphorylated RyR2 at serine 2808; SR: Sarcoplasmic Reticulum; SERCA: Sarcoplasmic Reticulum ATPase; STZ: Streptozotocin.

**Table 1 ijms-20-01680-t001:** Physical characteristics of the control, diabetes mellitus (DM), and empagliflozin-treated DM rats at 10 (baseline) and 16 weeks of age (after treatment).

Physical Characteristics	Control	DM	Empagliflozin-Treated DM
Baseline FBG (mM)	4.7 ± 0.2	4.7 ± 0.3	4.7 ± 0.1
FBG (mM) 2 weeks after streptozotocin	4.7	20.1 ± 0.7 ^a^	20.1 ± 0.9 ^a^
FBG (mM) after empagliflozin treatment	5.1 ± 0.1	20.7 ± 2.0 ^a^	10.0 ± 1.3 ^a,b^
Baseline SBP (mmHg)	110.1 ± 1.6	120.4 ± 4.4	114.6 ± 3.9
SBP (mmHg) after treatment	123.4 ± 7.6	122.7 ± 8.4	120.2 ± 7.2
Baseline DBP (mmHg)	64.2 ± 3.2	63.9 ± 3.1	62.2 ± 3.6
DBP (mmHg) after treatment	64.2 ± 3.2	64.6 ± 3.2	63.9 ± 5.5
Baseline HR (bpm)	414.4 ± 15.9	423.8 ± 14.5	421.8 ± 14.1
HR (bpm) after treatment	441.6 ± 10.8	345.5 ± 16.0 ^a^	394.4 ± 16 ^a,b^
Baseline BW (gm)	332.4 ± 4.8	323.7 ± 5	326.1 ± 7
BW (gm) after treatment	423.3 ± 10.3	278.2 ± 14.9 ^a^	296.2 ± 11.9 ^a,b^
HW (gm) after treatment	1.3 ± 0.0	1.5 ± 0.1	1.4 ± 0.1
HW/BW ratio (gm/kg) after treatment	3.2 ± 0.1	5.4 ± 0.3^a^	4.4 ± 0.1 ^a,b^

Abbreviations: FBG: Fasting Blood Glucose; SBP: Systolic Blood Pressure; DBP: Diastolic Blood Pressure; HR: Heart Rate; bpm: Beats Per Minute; BW: Body Weight; HW: Heart Weight. Values are expressed as the mean ± SEM; Number of rats: 7 per group. ^a^
*p* < 0.005 compared to the controls; ^b^
*p* < 0.05, *p* < 0.005 compared to the DM rats.

**Table 2 ijms-20-01680-t002:** Echocardiograms of control, diabetes mellitus (DM), and empagliflozin-treated DM rats at 16 weeks.

Group Studied	LVEDd (mm)	LVESd (mm)	EDV (mL)	ESV (mL)	EF (%)	FS (%)
Control	7.3 ± 0.2	3.5 ± 0.1	0.9 ± 0.1	0.1 ± 0.01	87.7 ± 0.7	53.4 ± 1.1
DM	8.3 ± 0.1 ^a^	4.7 ± 0.1 ^a^	1.2 ± 0.1 ^a^	0.3 ± 0.04 ^a^	79.0 ± 1.1 ^a^	42.9 ± 1.1 ^a^
Empagliflozin-treated DM	7.3 ± 0.1 ^b^	3.7 ± 0.1 ^b^	0.9 ± 0.0 ^b^	0.1 ± 0.04 ^b^	85.4 ± 1.5 ^b^	49.8 ± 1.7 ^b^

Abbreviations: LVEDd: Left Ventricular end-diastolic diameter; LVESd: Left Ventricular end-systolic diameter; EDV: End-diastolic Volume; ESV end-systolic volume; EF Ejection Fraction; FS: Fractional Shortening. Values are expressed as the mean ± SEM. Number of rats: 7 per group; ^a^
*p* < 0.005 compared to the control rats; ^b^
*p* < 0.005 compared to the DM rats.

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
