# Peer review of "Empagliflozin Attenuates Myocardial Sodium and Calcium Dysregulation and Reverses Cardiac Remodeling in Streptozotocin-Induced Diabetic Rats"

_ijms, 2019, doi:10.3390/ijms20071680_

Reviewer 1 Report

In the present manuscript, the authors investigated the effects of empagliflozin on cardiac dysfunction and calcium dysregulation in streptozotocin-induced diabetic (DM) model rats. As a result, empagliflozin ameliorated cardiac dysfunction as well as structural and electrical remodeling in DM rats through improving the decrease in calcium transient at least in part via the regulation of protein expression of ryanodine receptor (RyR) and sarcoplasmic reticulum calcium ATPase. While the authors clearly show the results, there are several points which should be revised and added in the present manuscript as below.

1.      In this study, the authors suggested the ‘direct’ effects of empagliflozin on streptozotocin-induced cardiac disorders. However, it is possible that hypoglycemic effects of empagliflozin led to the improvement of the cardiac condition. Thus, the authors should show other data which empagliflozin directly affects the DM rat heart.

2.      In Table 1, the body weight of DM rats was significantly decreased compared with that of control rats, leading to increased ratio of heart to body weight. If empagliflozin plays an inhibitory role against streptozotocin-induced cardiac hypertrophy, the authors should show other data including cross-sectional area of rat cardiomyocytes.

3.      Did the authors examine the changes in T-type calcium channels in DM and empagliflozin-treated DM rats?

4.      Action potential duration of repolarization is controlled not only by calcium or sodium channel but also by potassium channel. Did the authors examine the functional changes in potassium channel?

5.      In Fig. 5, ROS production in DM rat cardiomyocytes was significantly increased compared with that of control, which was inhibited by empagliflozin treatment. What is the mechanism of the ROS inhibition by empagliflozin? In addition, the relation between ROS production and the cardiac disorders is not clearly in this manuscript. The authors should detail that in Discussion section.

1.      The title is inadequate for this manuscript. It should be revised by using appropriate representation and word.

2.      In Fig. 5C, representative images between DM and empagliflozin-treated DM rat cardiomyocytes is seemed to be unchanged. Please replace it with appropriate images.

3.      In Fig. 6, while RyR2 phosphorylation at Ser2805 was significantly increased in DM rat heart compared with that of control, its expression level was conversely decreased. Why were the calcium sparks increased in DM rats?

4.      In Fig. 7, schematic diagram of this study is difficult to understand. In addition, site of action of empagliflozin should be clearly indicated in the schema.

5.      In discussion section, the authors should detail the action mechanisms of empagliflozin in this study.

Author Response

Dear Reviewer:

Thank you very much for your detailed comments. These comments were very instructive and very helpful to this manuscript and to our future research. The responses to these comments are enumerated below:                                                                                                  

1. In this study, the authors suggested the ‘direct’ effects of empagliflozin on streptozotocin-induced cardiac disorders. However, it is possible that hypoglycemic effects of empagliflozin led to the improvement of the cardiac condition. Thus, the authors should show other data which empagliflozin directly affects the DM rat heart.

-Thank you very much for this comment. We agree with your comment that it is possible that hypoglycemic effects of empagliflozin led to the improvement of the cardiac condition. Different from the effect of empagliflozin, our previous study has shown that the treatment of PPAR-g agonist, rosiglitazone in DM rats has prolonged APD with arrhythmogenic potential. These findings suggest that different DM medications may have dissimilar cardiac effects. Baartscheer et al. (Diabetologia. 2017; 60:568-573) have shown that empagliflozin directly affects DM rat heart. Therefore, empagliflozin may improve DM cardiomyopathy in addition to its hypoglycemic effects. However, this study did not measure the direct effects of empagliflozin on cardiomyocytes. According to your suggestions, we have deleted the comment of “direct effects” in the revised discussions and added this study limitation in the revised manuscript (page 9, lines 20-24; red fonts) as follows “First, we did not measure the direct effects of empagliflozin on cardiomyocytes, and it is possible that hypoglycemic effects of empagliflozin may contribute to the improvement of the cardiac condition. In this study, as compared to control, the empagliflozin-treated DM rats had hyperglycemia. Thus, empagliflozin may improve DM cardiomyopathy in addition to its hypoglycemic effects.”

2. Regarding the comment “In Table 1, the body weight of DM rats was significantly decreased compared with that of control rats, leading to increased ratio of heart to body weight. If empagliflozin plays an inhibitory role against streptozotocin-induced cardiac hypertrophy, the authors should show other data including cross-sectional area of rat cardiomyocytes.”

-We agree with your comment that we should show the data about the cross-sectional area of rat cardiomyocytes. According to your suggestions, we have measured the cross-sectional area of rat cardiomyocytes from different groups by confocal microscopy and found that the cell size in DM rat cardiomyocytes (3004±81 mm2, n=88) were larger than that in control (2801± 57 mm2, n=100) and empagliflozin-treated DM (2635±77 mm2, n=101) rat cardiomyocytes. This finding is compatible to the previous results done by Habibi et al. (Cardiovasc Diabetol. 2017; 16: 9). We have presented these data in the revised manuscript (page 4, lines 19-23; red font) as follows “Moreover, the measurement of the cross-sectional area in isolated single ventricular myocytes by confocal microscope showed a larger cell size in DM rat cardiomyocytes (3004±81 mm2, n=88) than in control (2801± 57 mm2, n=100) and empagliflozin-treated DM (2635±77 mm2, n=101) rat cardiomyocytes.” (page 6, lines 24-26; red font). “In this study, we found that the DM rats had a greater heart-to-body weight ratio and larger cross-sectional area of ventricular myocytes than control, and empagliflozin-treated groups. These findings were compatible with the results in the previous study (Habibi et al. Cardiovasc Diabetol. 2017;16: 9).

3. Regarding the comment “Did the authors examine the changes in T-type calcium channels in DM and empagliflozin-treated DM rats? ”

-We appreciated this comment very much. We understand your concern about the changes in T-type calcium channels in DM and empagliflozin-treated DM rats. In the adult heart, the T-type Ca2+ channel is almost undetectable in ventricular myocytes, and is re-expressed in ventricular myocytes under various pathological conditions such as hypertrophy. However, this study did not measure T-type Ca2+ channel since it provides a smaller contribution to the trigger for Ca2+ release than does the L-type Ca2+ channel (Ono K. J Mol Cell Cardiol. 2010;48:65-70.). So we did not study the role of T-type Ca2+ channel in our animals. According to your suggestions, we have provided this limitation in the revised manuscript (page 9, lines 24-27; red font) as follows “Second, T-type Ca2+ channel has been shown to be re-expressed in the hypertrophic ventricular myocytes (Ono K et al. J Mol Cell Cardiol. 2010;48:65-70.) However, this study did not measure T-type Ca2+ channel because T-type Ca2+ channel contributes less to the trigger for Ca2+ release (Ono K et al. J Mol Cell Cardiol. 2010;48:65-70).”

4. Regarding the comment “Action potential duration of repolarization is controlled not only by calcium or sodium channel but also by potassium channel. Did the authors examine the functional changes in potassium channel?”

-We appreciated this comment very much. In the present study, we focused on the effect of empagliflozin on calcium/sodium regulation in DM cardiomyocytes, thus we did not study the role of potassium channel in action potential morphology. According to your suggestions, we have added this limitation in the revised manuscript (page 9, lines 27-30; red font) as follows “Additionally, APD is controlled not only by Ca2+ or Na+ channel but also by potassium (K+) channel, but this study mainly targeted the effects of empagliflozin on Ca2+/Na+ regulation in DM cardiomyocytes. The functional changes in K+ channels were not examined.”

5. Regarding the comment about “In Fig. 5, ROS production in DM rat cardiomyocytes was significantly increased compared with that of control, which was inhibited by empagliflozin treatment. What is the mechanism of the ROS inhibition by empagliflozin? In addition, the relation between ROS production and the cardiac disorders is not clearly in this manuscript. The authors should detail that in Discussion section.” 

-Thank you very much for this comment. Previous study has shown that local Ca2+ and ROS cross-signaling between SR and mitochondria. The RyR-dependent SR Ca2+ release contributes to mitochondrial Ca2+ overload, which facilitates ROS production. (Csordás G et al. Biochim Biophys Acta. 2009;1787:1352-62.) Mitochondria-derived ROS could induce local ER Ca2+ release events, which increased spark frequency in cardiomyocytes (S.M. Davidson, M.R. Duchen. Cell Calcium. 2006;40:561-574). In our study, the interruption of damage feedback loop by empagliflozin may be the mechanism of the ROS inhibition by empagliflozin. According to your suggestion, we have discussed this issue in the revised manuscript (page 8, lines 29-35; red font) as follows: “Ca2+ sparks induced by RyR hyperphosphorylation lead to mitochondrial Ca2+ overload, facilitating ROS production. In addition, mitochondria-derived ROS could induce local ER Ca2+ release events, which increased spark frequency in the cardiomyocytes. The reduction of Ca2+ sparks by empagliflozin may contribute to its inhibitory effect on ROS production. (S.M. Davidson et al. Cell Calcium. 2006;40:561-574). Moreover, the hypoglycemic effects of empagliflozin may reduce ROS since hyperglycemia and insulin resistance result in excess ROS production (Teshima Y et al. Circ J. 2014;78:300-306).”

6. Regarding the comment about “The title is inadequate for this manuscript. It should be revised by using appropriate representation and word.”  

-We appreciate this comment very much. According to your suggestion, we have changed the title of this manuscript as “Empagliflozin Attenuates Myocardial Sodium and Calcium Dysregulation and Reverses Cardiac Remodeling in Streptozotocin-induced Diabetic Rats

7. Regarding the comment about “In Fig. 5C, representative images between DM and empagliflozin-treated DM rat cardiomyocytes is seemed to be unchanged. Please replace it with appropriate images.”

-We apologized for the poor images provided in Fig. 5C. According to your suggestion, we have replaced it with an appropriate image.

8. Regarding the comment “In Fig. 6, while RyR2 phosphorylation at Ser2805 was significantly increased in DM rat heart compared with that of control, its expression level was conversely decreased. Why were the calcium sparks increased in DM rats?”

-Thank you very much for this comment. We understand your concern about the converse had been shown in RyR expression and phosphorylation. Similar to the previous study (Yaras N et al. AJP Heart and Circulatory Physiology. 2007; 292:H912-20), streptozotocin-treated rats had lower expression of RyR and higher RyR2 phosphorylation at Ser2808 than control, which may be caused by hyperphosphorylation in DM cardiomyopathy and represents alteration in cardiac RyR function. Hyperphosphorylated RyR increases RyR opening probability, and a small SR Ca2+ leak via RyR leads to the genesis of Ca2+ spark. According to your suggestions, we discussed this point in the revised manuscript (page 8, lines 8-13; red font) as follows “Similar to the previous study (Yaras N et al. AJP Heart and Circulatory Physiology. 2007; 292:H912-20), streptozotocin (STZ)-treated rats had lower expression of RyR and higher RyR2 phosphorylation at Ser2808 than control, which may be caused by hyperphosphorylation in DM cardiomyopathy and represents alteration in cardiac RyR function. Hyperphosphorylated RyR increases RyR opening probability, and a small SR Ca2+ leak via RyR leads to the genesis of Ca2+ spark.”

9. Regarding the comment “In Fig. 7, schematic diagram of this study is difficult to understand. In addition, site of action of empagliflozin should be clearly indicated in the schema.”

- We apologized for the difficulty in understanding of our schematic diagram. According to your suggestion, we have modified the schema to make it understandable. We have indicated the site of action of empagliflozin in the schema.

10. Regarding the comment “In discussion section, the authors should detail the action mechanisms of empagliflozin in this study.”

-Thank you very much for this comment. According to your suggestions we have detailed the mechanism of empagliflozin in this study at the end of our discussion (page 9, lines 12-17; red font) as follows: “Figure 7 detailed the potential action mechanisms of empagliflozin in DM hearts in this study. The treatment of empagliflozin may reverse DM-induced Ca2+/ Na+ dysregulation through decrease of ROS and ionic channel modification in cardiomyocytes, leading to the improvement of cardiac function, attenuation of ventricular hypertrophy, and correction of prolonged QT interval.”

The above descriptions are the responses to your comments and suggestions. We highly appreciated your careful and detailed review on our manuscript. Thank you very much.

Sincerely yours,

Yi-Jen Chen, MD, PhD

Director, Cardiovascular Research Center, Wan Fang Hospital, Taipei Medical University; Professor, Graduate Institute of Clinical Medicine, Taipei Medical University

Reviewer 2 Report

Cople of questions to authors. 

First, why the DM rats had a lower heart rate compared to the control and empagliflozin‐treated rats after treatmet? In people the DM patients have a greater heart rate than nonDM persones. 

Second one, why the lower body weights after treatment were noted  in the DM and empagliflozin‐treated DM groups? Usually DM subjects have greater body weights than nonDM and after the treatment it falls down. In rat model we notes inverse situation than in people. Thees facts need to explane in the discussion. 

Author Response

Dear Reviewer:

Thank you very much for your detailed comments. These comments were very instructive and very helpful to this manuscript and to our future research. The responses to these comments are enumerated below:

1.      Regarding the comment “First, why the DM rats had a lower heart rate compared to the control and empagliflozin‐treated rats after treatment? In people the DM patients have a greater heart rate than nonDM persons”

-Thank you very much for the comment. In this study, similar to those in the previous rodent studies, we also noted a lower heart rate in STZ-induced DM rat model. (Lee TI et al. Int J Cardiol. 2013;165: 299-307; Malone JI Malone et al. Cardiovas Diabetol. 2006; 5:2; Howarth FC et al. Exp Physiol. 2005;90:827-835). Increased resting heart rate is found more in DM than nonDM people (Hossein F et al. ISRN Endocrinology. 2012; Volume 2012: Article ID 168264), which may be caused by diabetic cardiovascular autonomic neuropathy (Freccero C et al. Diabetes Care. 2004;27:2936–2941). We have discussed this discrepancy between the DM patients and DM animals in the revised limitation (page 9, lines 30-33; red font) as follows: “Finally, the experimental setting in this study may not fully correlate with clinical features. Most DM patients have faster resting heart rate due to autonomic neuropathy (Hossein F et al. ISRN Endocrinology. 2012; Volume 2012: Article ID 168264; C. Freccero C et al. Diabetes Care. 2004;27: 2936-2941). Differently, this study found that DM rats had lower heart rates compared to the control.”

2. Regarding the comment “Second one, why the lower body weights after treatment were noted in the DM and empagliflozin‐treated DM groups? Usually DM subjects have greater body weights than nonDM and after the treatment it falls down. In rat model we notes inverse situation than in people. These facts need to explain in the discussion.”

-We appreciate your comment very much. We understand your concern about the difference of our rat body weight with DM patients. It is true that most patients with type 2 DM are obese (Eckel RH et al. J Clin Endocrinol Metab. 2011;96:1654) and there is a strong association between obesity and type 2 DM (Guh, DP et al. BMC Public Health. 2009;9:88). However, our model is more of a type 1 DM model, thus similar to previous rodent studies, our rat has lower body weight after streptozocin-induced DM (Lee TI et al, Int J Cardiol. 2013;165:299-307; Zafar M et al. Int. J. Morphol. 2010;28:135-142; Howarth FC et al. Exp Physiol. 2005;90: 827-835). In addition, the lower body weight in empagliflozin-treated DM rats may be caused by their hyperglycemia, which is incompletely controlled by empagliflozin. According to your suggestions, we have added this in the discussion of our revised manuscript (page 9, lines 34-38; red font) as follows: “STZ-induced DM resembles human type 1 DM, which is associated with lower body weight. Nevertheless, most DM patients have greater body weights than non-DM subjects, since type 2 DM is more common. This study showed that treatment of empagliflozin did not reverse the lower body weight in DM rats, which may be caused by inadequate glycemic control.”

The above descriptions are the responses to your comments and suggestions. We highly appreciated your careful and detailed review on our manuscript. Thank you very much.

Sincerely yours,

Yi-Jen Chen, MD, PhD

Director, Cardiovascular Research Center, Wan Fang Hospital, Taipei Medical University; Professor, Graduate Institute of Clinical Medicine, Taipei Medical University

Round  2

Reviewer 1 Report

The authors have revised the original manuscript in response to my previous comments. However, the answers are imperfectly. Please revise in accordance with the comments below.

1.       EMPA-REG outcome suggested in the patients with type 2 diabetes that empagliflozin significantly decreased death from cardiovascular causes and hospitalization for heart failure (N Engl J Med, 2015). However, it is unknown whether the effects of empagliflozin on diabetic cardiomyopathy is directly or indirectly. While the authors tested the hypothesis that empagliflozin directly affects cardiac disorders in diabetic model rats, it cannot be clearly proved in the authors’ present study. At least, examination of acute effects of empagliflozin on electromechanical function in normal or diabetic rat cardiomyocytes are needed. This is one of the most important things in this study.

2.       The authors should add the levels of fasting blood glucose two weeks after streptozotocin injection (before treatment with empagliflozin) to Table 1.

3.       The results of the cross-sectional area in isolated cardiomyocytes should be mentioned in Result 2.1 but not 2.2. In addition, the procedure and analysis are needed in the Methods section.

4.       In results section, please insert the measured value (SEM), number of sample and P value.

5.       In each title of the results, ‘Effects of DM and empagliflozin on …’ should be replaced by ‘Effects of empagliflozin on … in DM rats (or streptozotocin-induced …)’.

6.       In new Fig. 7, what does ‘- Increase left ventricular end-diastolic diameter’ mean? ‘Cardiac dysfunction and remodeling’ is better.

7.       Please replace the following words:

Line 30; ‘Electrocardiographic’ to ‘Electrocardiography’

Line 30; ‘echocardiographic’ to ‘echocardiography’

Line 31; ‘confocal microscopy’ to ‘confocal microscopic examination

8.       In line 30, please delete ‘fluo-3 fluorometric ratio’. Confocal microscopic examination includes it.

9.       In line 44 (keywords), ‘empagliflozin’ should be deleted. And line 45 (keywords), since ‘sodium’ is inadequate, please replace with another word.

10.   This manuscript should get some English proofreading.

Author Response

Responses to Reviewer #1

Dear Reviewer:

Thank you very much for your detailed comments. These comments were very instructive and very helpful to this manuscript and to our future research. The responses to these comments are enumerated below:

1. EMPA-REG outcome suggested in the patients with type 2 diabetes that empagliflozin significantly decreased death from cardiovascular causes and hospitalization for heart failure (N Engl J Med, 2015). However, it is unknown whether the effects of empagliflozin on diabetic cardiomyopathy is directly or indirectly. While the authors tested the hypothesis that empagliflozin directly affects cardiac disorders in diabetic model rats, it cannot be clearly proved in the authors’ present study. At least, examination of acute effects of empagliflozin on electromechanical function in normal or diabetic rat cardiomyocytes are needed. This is one of the most important things in this study

- We agree with your comment that we should at least performed an experiment on the acute effect of empagliflozin electromechanical function in normal or diabetic rat cardiomyocytes. According to your suggestion, we performed the acute effect of empagliflozin (1 μM EMPA) on diabetic rats. We have found that acute administration of empagliflozin (1 mM) reduced the APD90 (from 170±4 ms to 129±11 ms, n=10, p<0.01), and APD50 (from 77±6 ms to 60±6 ms, n=10, p<0.05) in DM ventricular myocytes. In addition, empagliflozin (1 mM) reduced INa-Late from 0.66±0.07 pA/pF to 0.28±0.04 pA/pF (n=11, p<0.005) and cytosolic ROS levels from 108±4 F/F0, to 81±4 F/F0 (n=11, p<0.005 ) in DM ventricular myocytes.” These findings suggest that empagliflozin may have direct effects on cardiac disorder in DM rats. We have added these findings in the revised results (page 6, lines 21-28, red font) and revised Figure 6 as follows “In order to study whether empagliflozin may directly affect cardiac disorder in DM rats, we investigated the acute effects of empagliflozin (1 mM) on isolated DM rat cardiomyocytes. As shown in Figure 6, acute administration of empagliflozin (1 mM) reduced the APD90 (from 170±4.0 ms to 129±11 ms, n=10, p<0.01), and APD50 (from 77±6 ms to 60±6 ms, n=10, p<0.05) in DM ventricular myocytes. In addition, empagliflozin (1 mM) reduced INa-Late from 0.66±0.07 pA/pF to 0.28±0.04 pA/pF (n=11, p<0.005) and cytosolic ROS levels from 108.0±4.0 F/F0, to 81±4 F/F0 (n=11, p<0.005 ) in DM ventricular myocytes.” We also have discussed these findings in the revised discussions (page 9, lines 16-23, red font) as follows “Our study also found that acute administration of empagliflozin (1 mM) reduced APD, INa-late, and ROS in DM ventricular myocytes, which was similar to the results seen from the in vivo treatment of empagliflozin in DM rat model. Previous study has shown that patients have peak plasma concentration of empagliflozin close to 1 mM (Laffell LMD, et al Diabet Med. 2018; 35:1096-1104), thus the concentration of empagliflozin used in this study is clinically relevant. These findings suggest that empagliflozin may have direct effects on cardiac disorder in DM rats. However, the hypoglycemic effects of empagliflozin may have also contributed to the improvements in the cardiac condition in DM cardiomyopathy.”

2. Regarding the comment “The authors should add the levels of fasting blood glucose two weeks after streptozotocin injection (before treatment with empagliflozin) to Table 1.

-Thank you very much for this comment. According to your suggestion, we added the fasting blood glucose after streptozotocin injection in the revised Table 1. (page 25, line 5, red font).

3. Regarding the comment “The results of the cross-sectional area in isolated cardiomyocytes should be mentioned in Result 2.1 but not 2.2. In addition, the procedure and analysis are needed in the Methods section. ”

-We appreciate this comment very much. According to your suggestion, the results of the cross-sectional area in isolated cardiomyocytes was moved from Result 2.2 to Result 2.1 (page 4, lines 20-23, red font) as follows “Moreover, measurements of the cross-sectional area in isolated single ventricular myocytes in confocal microscopic examinations showed a larger cell size in the DM group (3004±81 mm2, n=88) than in the control (2801± 57 mm2, n=100, p<0.05) and empagliflozin-treated DM (2635±77 mm2, n=101, p<0.005) groups. In addition, we added the procedure and analysis in the Methods section (page 14, lines, 4-6, red font) as follows: “The cross-sectional area in the isolated single ventricular myocytes were imaged using a confocal laser scan microscope (Leica Microsystems) and images were processed by ImageJ measurement tools (Blackwell DJ et al. Circ Res. 2017; 121:1323-1330).”

4. Regarding the comment “In results section, please insert the measured value (SEM), number of sample and P value.”

-We appreciate this comment very much. According to your suggestions, we have inserted the measured value (SEM), number of sample and P values in the results section in red font.

5. Regarding the comment about “In each title of the results, ‘Effects of DM and empagliflozin on …’ should be replaced by ‘Effects of empagliflozin on … in DM rats (or streptozotocin-induced …).”

-Thank you very much for this comment. According to your suggestions, the title of the results were replaced as follows: “2.2 Effects of empagliflozin on the action potentials (APs) in DM rats (page 4, line 24, red font); 2.3 Effects of empagliflozin on the Ca2+ stores in DM rats (page 4, line 32, red font); 2.4 Effects of empagliflozin on L-type Ca2+ channel (ICa-L) current and NCX current in DM rats (page 5, lines 22-23, red font); 2.5 Effects of empagliflozin on late Na+ channel (INa-late) current and Na+/hydrogen (Na+/H+) exchanger current in DM rats  (page 5, line 30-31, red font); 2.6 Effects of empagliflozin on oxidative stress in DM rats (page 6, line 5, red font); 2.8 Effects of empagliflozin on Ca2+ regulatory proteins in DM rats (page 6, line 28, red font)”

6. Regarding the comment about “In new Fig. 7, what does ‘- Increase left ventricular end-diastolic diameter’ mean? ‘Cardiac dysfunction and remodeling’ is better.”

-We appreciate this comment very much. According to your suggestion, we have changed Increase left ventricular end-diastolic diameter to Cardiac dysfunction and remodelling in the revised Figure 8.

7. Regarding the comment about “Please replace the following words: Line 30; ‘Electrocardiographic’ to ‘Electrocardiography’ Line 30; ‘echocardiographic’ to ‘echocardiography’ Line 31; ‘confocal microscopy’ to ‘confocal microscopic examination”

- Thank you very much for this comment. According to your suggestion, we have replace the following words: ‘Electrocardiographic’ to ‘Electrocardiography’ (Page 11, line 16, red font); ‘echocardiographic’ to ‘echocardiography’ (Page 11, line 16, red font); ‘confocal microscopy’ to ‘confocal microscopic examination (Page 12, lines 6-7, red font).

8. Regarding the comment “In line 30, please delete ‘fluo-3 fluorometric ratio’. Confocal microscopic examination includes it. ”

-Thank you very much for this comment. According to your suggestion, we deleted ‘fluo-3 fluorometric ratio’ (page 11, line 35).

9. Regarding the comment “In line 44 (keywords), ‘empagliflozin’ should be deleted. And line 45 (keywords), since ‘sodium’ is inadequate, please replace with another word.

- We appreciate this comment very much. According to your suggestion we deleted ‘empagliflozin’, and replace sodium to “sodium regulation” (page 2, line 27).

10. Regarding the comment “This manuscript should get some English proofreading.”

-Thank you very much for this comment. According to your suggestion, we have send our manuscript to a native English speaker for proof reading.

The above descriptions are the responses to your comments and suggestions. We highly appreciated your careful and detailed review on our manuscript. Thank you very much.

Sincerely yours,

Yi-Jen Chen, MD, PhD

Director, Cardiovascular Research Center, Wan Fang Hospital, Taipei Medical University; Professor, Graduate Institute of Clinical Medicine, Taipei Medical University

Round  3

Reviewer 1 Report

Please correct the following points;

Line 60: hearth to heart

The sample number in DM rats is different between Result 2.3. (Line 139) and Figure 2B. Please confirm the number of sample in other experiments.

There is no change in new Fig. 8 compared with previous version (Fig. 7 in revised version 1).

Author Response

Responses to Reviewer #1

Dear Reviewer:

Thank you very much for your detailed comments. These comments were very instructive and very helpful to this manuscript and to our future research. The responses to these comments are enumerated below:

1. Regarding the comment “Line 60: hearth to heart”

- We apologize for the typographical error made. We have replaced the “hearth” to “heart” in the introduction (page 3, line 15; red font).

2. Regarding the comment “The sample number in DM rats is different between Result 2.3. (Line 139) and Figure 2B. Please confirm the number of sample in other experiments.

-Thank you very much for this comment. We have rechecked the number of sample in Figure 2B. We apologize for the typing error in Result 2.3, the number of sample should be 13 (page 5, line 9; red font) similar to Figure 2B.  

3. Regarding the comment “There is no change in new Fig. 8 compared with previous version (Fig. 7 in revised version 1).”

-We are very sorry for sending the previous version. We have replaced the new version of Figure 8.

The above descriptions are the responses to your comments and suggestions. We highly appreciated your careful and detailed review on our manuscript. Thank you very much.

Sincerely yours,

Yi-Jen Chen, MD, PhD

Director, Cardiovascular Research Center, Wan Fang Hospital, Taipei Medical University; Professor, Graduate Institute of Clinical Medicine, Taipei Medical University
